# M$^2$Hub: Unlocking the Potential of Machine Learning for Materials Discovery

**Yuanqi Du**[1,*]    **Yingheng Wang**[1,*]    **Yining Huang**[2]    **Jianan Canal Li**[3]    **Yanqiao Zhu**[4]
**Tian Xie**[5]    **Chenru Duan**[6]    **John M. Gregoire**[7]    **Carla P. Gomes**[1]
[1] Cornell    [2] Northwestern    [3] UCB    [4] UCLA
[5] MSR AI4Science    [6] Microsoft Quantum    [7] Caltech    [*] Equal Contribution

## Abstract

We introduce M$^2$Hub, a toolkit for advancing machine learning in materials discovery. Machine learning has achieved remarkable progress in modeling molecular structures, especially biomolecules for drug discovery. However, the development of machine learning approaches for modeling materials structures lag behind, which is partly due to the lack of an integrated platform that enables access to diverse tasks for materials discovery. To bridge this gap, M$^2$Hub will enable easy access to materials discovery tasks, datasets, machine learning methods, evaluations, and benchmark results that cover the entire workflow. Specifically, the first release of M$^2$Hub focuses on three key stages in materials discovery: virtual screening, inverse design, and molecular simulation, including 9 datasets that covers 6 types of materials with 56 tasks across 8 types of material properties. We further provide 2 synthetic datasets for the purpose of generative tasks on materials. In addition to random data splits, we also provide 3 additional data partitions to reflect the real-world materials discovery scenarios. State-of-the-art machine learning methods (including those are suitable for materials structures but never compared in the literature) are benchmarked on representative tasks. Our codes and library are publicly available at `https://github.com/yuanqidu/M2Hub`.

## 1  Introduction

With the methodological advancements in machine learning, an increasing number of machine learning models have been developed and applied to solve scientific problems, from simulating molecular systems with millions of particles to predicting accurate protein structures Zhang et al. [2018], Jumper et al. [2021]. The primary focus of machine learning in the chemical sciences has remained in the domain of molecular structures, (bio)molecules including small molecules, proteins, RNAs, etc. Atz et al. [2021], Rives et al. [2021], Townshend et al. [2021]. However, materials constitute a large portion of the chemical space which have been significantly less studied, especially in the machine learning community. Among scientific problems, materials discovery plays a vital role in driving innovations and progress across various fields spanning energy, electronics, healthcare, and sustainability Sanchez-Lengeling and Aspuru-Guzik [2018], Gomes et al. [2021]. However, the traditional trial-and-error approach to materials discovery is expensive and time-consuming. Over decades, classical machine learning methods have already been widely applied in assisting materials discovery, Schmidt et al. [2019] yet the impact of machine learning for solid state materials lags behind its efficacy in other areas of chemical science.

Witnessing the success of machine learning in solving grand challenges in science Wang et al. [2018], Jumper et al. [2021], one of the key ingredients is the infrastructure that supports the machine learning community to build the machine learning workflow: data preparation/processing, model development, performance evaluation, and model improvement based on the evaluation feedback. While effort

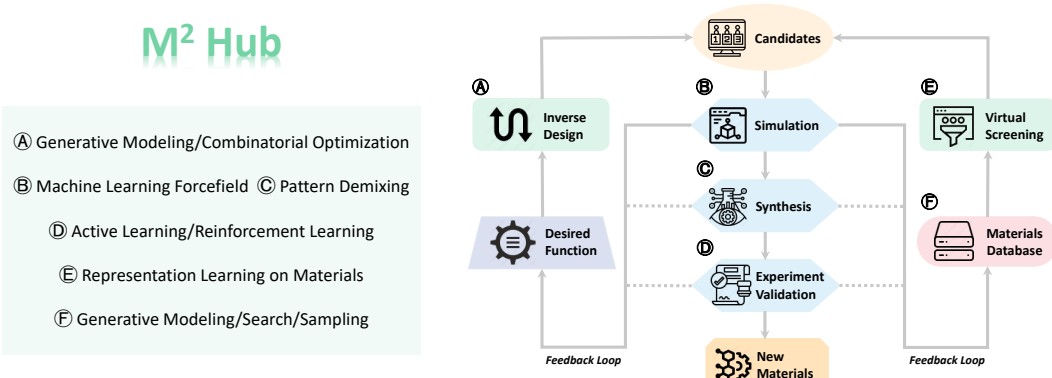

Figure 1: M²Hub: Materials discovery meets Machine learning. A-F on the left figure demonstrates machine learning approaches used in each stage of the materials discovery pipeline on the right figure (dashed lines denote currently unavailable experiment-related tasks).

has been made to make materials datasets available to the machine learning community Blaiszik et al. [2019], Dunn et al. [2020], Clement et al. [2020], Qayyum et al. [2022], Durdy et al. [2023], existing work mainly focus on providing data servers that allow the users to query materials data and predefined benchmark sets. However, to bridge the gap between the molecular and solid state materials, we identify the need for a unified platform to facilitate the development of machine learning approaches for materials discovery purpose, including (1) centralized data sources with diverse materials, property and task types, (2) clear problem formulations, (3) realistic problem settings (e.g. data split), and (4) appropriate benchmarks and transparent comparisons with prior methods.

In order to address the aforementioned challenges, we establish M²Hub, which integrates and connects different machine learning building blocks in the materials discovery (Fig. 1). The cornerstone of M²Hub is a benchmark that incorporates several key aspects: (i) it integrates three key tasks: virtual screening, molecular dynamics simulation, and inverse design, which are translated using machine learning constructs such as materials representation learning, machine learning forcefield, and generative materials design; (ii) it is underpinned by a curated set of 11 datasets that encompass 6 types of materials with 56 tasks across 8 distinct material properties. In addition to the standard random split, we have included 3 realistic (out-of-distribution) data splits to enhance the robustness of model evaluation; (iii) a distinctive feature of our benchmark is the emphasis on the generative design of materials. For this, we provide machine learning formulations, evaluation metrics, and oracle functions to facilitate further research and development in this area; (iv) finally, our benchmarks evaluate not only the commonly used material representation learning methods, but also those designed for non-periodic molecular structures. These methods are applied to 13 representative tasks for material property prediction.

The flow of this paper is as follows: we introduce the background and related work on developing machine learning methods for materials discovery in Sec. 2; we present the overview of M²Hub including problem formulation, dataset curation, data processing, evaluation and oracle function for inverse design in Sec. 3; in Sec. 4, we detail the implemented machine learning models, benchmarking results, observations, and insights emerging from the results.

## 2 Background

### 2.1 Materials Representation Learning

Material representation learning refers to representing materials structures in an expressive and machine-readable format for downstream studies, from property prediction to materials generation. Recent advances in graph neural networks bring a wave of representing materials structures as graphs where nodes represent atoms and edges represent bonds or interactions. A line of works has been proposed to adapt this structured inductive bias into deep learning models.CGCNN Xie and Grossman [2018] introduces a multi-edge graph representation to capture periodicity. MEGNet Chen et al.

[2019] unifies molecules and crystal structures representations by graph neural networks representing each atom as node and interaction/bond between atoms as edge. More recently, ALIGNN considers both atomistic and line graphs which externally capture angular information Choudhary and DeCost [2021]. Equivariant graph neural networks (e.g., E3NN Geiger and Smidt [2022]) have also been applied thanks to its roto-translational equivariance property. This task has also attracted increasing interests through years Kaba and Ravanbakhsh [2022], Yan et al. [2022], DAS et al. [2023], Lin et al. [2023], Das et al. [2023].

## 2.2 Machine Learning Forcefields

Molecular dynamics simulation has become an essential tool to understand the microscopic dynamical behaviors of molecular systems. It is worth noting there is a common trade-off between two popular diagrams, empirical forcefields and *ab initio* molecular dynamics. Empirical forcefields often rely on the hand-crafted parameters which are efficient yet inaccurate, while ab initio molecular dynamics rely on quantum-mechanical calculations which are precise but inefficient. Inspired by recent advances of deep learning in automated parameters learning and transferability, a large amount of works has been developed to learn machine learning forcefields from quantum-mechanical data to strike a balance between accuracy and efficiency. Specifically, it is expected to be more accurate than empirical forcefields and more efficient than quantum-mechanical calculations. Most representative work include DeepMD Zhang et al. [2018], ANI-1 Smith et al. [2017], and NeuqIP Batzner et al. [2022].

## 2.3 Materials Inverse Design

Designing new materials structures is a long-standing challenge, often known as the inverse design problem, in materials science Du et al. [2022a], Manica et al. [2023]. Before deep generative models have been applied to this problem, traditional computational methods often leverage quantum mechanical search over the possible stable materials including random search, evolutionary algorithm, element substitutions over known materials Glass et al. [2006], Pickard and Needs [2011], Hautier et al. [2011]. One line of works leverages a learned force field to minimize the energy of the structure to reach a stable material Deringer et al. [2018]. Later, deep generative models have been applied to this problem where the models aim to model the distribution of the known crystal structures and learn to sample new structures from it. Early work leverage 3D voxel representation but it is nontrivial to fit atom from the generated voxels Hoffmann et al. [2019], Noh et al. [2019]. Later work instead leverage atomic representation directly Zhao et al. [2021]. G-SchNet Gebauer et al. [2019] instead proposes an auto-regressive model that generates each atom in a sequential way. Notably, it remains largely unexplored for efficient and controllable crystal structure generation with machine learning methods. A recent work Xie et al. [2022] builds upon the recent success of diffusion models on images and adapts them for crystal structure generation and optimization in an iterative refinement manner instead of one-shot or auto-regressive sequential generation.

# 3 Overview of M²Hub

## 3.1 Problem Formulation

**Material representation.** Material structures can be represented as a set of atoms $M = (m_0, m_1, \ldots, m_N)$ in 3D space with atom types $H = (h_0, h_1, \ldots, h_N) \in \mathbb{R}^{N \times K}$ and atomic positions $X = (x_0, x_1, \ldots, x_N) \in \mathbb{R}^{N \times 3}$ where $N$ denotes number of atoms and $K$ denotes number of atom types (C, O, Fe, Al, etc.). Most materials are crystal structures which periodically repeat their unit cells in 3D space. In such cases, lattice vectors $L = (l_1, l_2, l_3) \in \mathbb{R}^{3 \times 3}$ are utilized to describe the periodicity in 3D space. Note $L$ is not rotation invariant, 6 invariant lattice parameters (lengths of lattice parameters and angles between them) can also

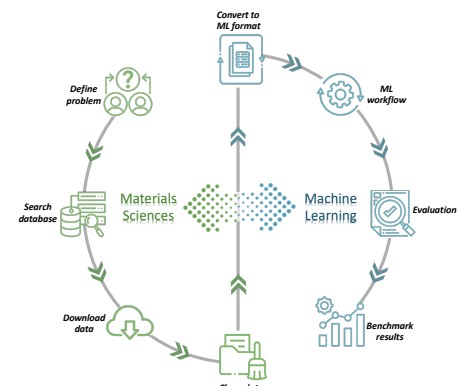

Figure 2: Regular workflow for studying materials with machine learning approaches (green colored steps denote materials science expertise and blue colored steps denote machine learning expertise.

Table 1: Curated materials datasets in M²Hub. Materials type and property type are detailed in Sec. 3.2.1 and Sec. 3.2.2; dim refers to data dimensionality; PBC refers to perodic boundary condition; method refers to how properties are obtained (sim short for simulation).

| Data name | Materials type | Dim | PBC | Prop. type | Task type | # tasks | # data | Size | Method |
|---|---|---|---|---|---|---|---|---|---|
| MatBench | inorganic bulk | 3D | (T, T, T) | elec./mech./stab./opt/ther. | scalar | 8 | 312–132,752 | 27.08±28.27 | Sim. |
| QMOF | metal-organic framework | 3D | (T, T, T) | elec. | scalar | 1 | >20,000 | 113.67±68.86 | Sim. |
| OC20 | bulk-adsorbate interface | 3D | (T, T, F) | energetic | scalar/vector | 3 | 640,081 | 73.26±30.96 | Sim. |
| OMDB | organic crystal | 3D | (T, T, T) | elec. | scalar | 1 | 12,500 | 82.29±26.55 | Sim. |
| DFT3D | inorganic bulk | 3D | (T, T, T) | elec./mech./stab./semi. | scalar | 29 | 55,722 | 9.59±8.04 | Sim. |
| DFT2D | inorganic bulk | 3D | (T, T, F) | stab. | scalar | 1 | 636 | 7.19±4.35 | Sim. |
| EDOS-PDOS | inorganic bulk | 3D | (T, T, T) | elec./ther. | 1D dist. | 2 | 48,469 | 9.50±8.45 | Sim. |
| tmQM | transition metal complex | 3D | (F, F, F) | elec. | scalar | 8 | 86,665 | 65.99±27.01 | Sim. |
| QM9 | organic molecules | 3D | (F, F, F) | elec. | scalar | 12 | ~134,000 | 18.02±2.94 | Sim. |
| Carbon24 | inorganic bulk | 3D | (T, T, T) | N/A | N/A | 1 | 10,153 | 9.21±3.58 | N/A |
| Perov5 | inorganic bulk | 3D | (T, T, T) | N/A | N/A | 1 | 18,928 | 5.00±0.00 | N/A |

be used to describe the lattice $(l_a, l_b, l_c, \alpha, \beta, \gamma)$. Overall, a material is denoted as $M = (H, X, L)$ if it is periodic and otherwise $M = (H, X)$.

**Material graph representation.** Regardless of periodicity, materials structures can be naturally represented as graphs $G = (\mathcal{V}, \mathcal{E})$, where $\mathcal{V}$ is a set of vertices and $\mathcal{E} = \{e_{ij}(k_1, k_2, k_3) | i, j \in \{1, 2, \ldots, N\}, k_1, k_2, k_3 \in \mathbb{Z}\}$ is a set of $D$ edges ($k_1, k_2, k_3$ denotes the translation of the unit cell using lattice vector $L$, none if not periodic); $H \in \mathbb{R}^{N \times K}$ denotes the node features; $X^{N \times 3}$ denotes the atomic positions; $E \in \mathbb{R}^{D \times F}$ denotes $F$ edge features (such as bonds or distances between each pair of nodes). The graph connections can be determined in multiple ways such as distance threshold, detailed in Sec. 3.3.

**Predictive tasks.** Predictive tasks often have paired input material $M$ and label $Y$, where given an input material $M$, we aim to predict the expected label as $p(Y|M)$. The label $Y$ could be of various format such as binary, scalar, vector and distribution, detailed in Sec. 3.4. Both material representation learning and machine learning forcefield are considered as predictive tasks.

**Generative tasks.** Generative tasks could be divided into two parts: (1) distribution learning and (2) goal-oriented generation. Given a set of $J$ materials $\mathcal{M} = \{M_i\}_{i=1}^{J}$, distribution learning aims to learn the distribution of $p(M)$ and sample new materials $M_{\text{new}} \sim p(M)$. Goal-oriented generation aims to sample molecules fulfilling specific design targets (often defined by an oracle function $f(M)$ such that $M^\star = \arg\max_{M \sim p(M)} f(M)$.

### 3.2 Data Description

We aim to curate a set of datasets covering the diversity of material, property, and task types and data amounts which will enable a variety of perspectives for machine learning model developments.

#### 3.2.1 Material types

**Inorganic bulks** refer to solid substances that lacks carbon–hydrogen bonds, that is, substances that are not organic compounds, such as metals, alloys, ceramics, and minerals. They are typically large-scale structures and are used in various applications, ranging from construction materials to electronics, due to their robustness and electrical/thermal conductivity.

**Organic crystals** are composed of carbon-based molecules arranged in a highly ordered pattern. They exhibit distinct molecular structures and are often used in the field of optoelectronics, such as organic light-emitting diodes (OLEDs), due to their unique optical properties. They can range in size from microscopic to macroscopic.

**Organic molecules** are individual carbon-based compounds that can be small in size, consisting of a few atoms, or large, such as polymers. They have diverse chemical structures and are widely used in pharmaceuticals, plastics, and organic electronics due to their flexibility in design and functionality.

**Bulk-adsorbate interfaces** refer to the boundary between a bulk material and an absorbed species, such as gases or liquids, on its surface. They are typically at the nanoscale and play a crucial role in various fields, including catalysis, gas sensing, transport, and energy storage, by influencing the interaction and reactivity of the absorbed species with the material.

**Transition metal complexes** are coordination compounds consisting of central transition metal atom(s) surrounded by ligands. They exhibit unique electronic and magnetic properties and are commonly used in catalysis, medicine, and material science due to their ability to undergo redox reactions and flexible and tunable coordination environment with organic ligands.

**Metal-organic frameworks (MOFs)** are crystalline materials composed of metal ions or clusters coordinated with organic linkers. MOFs poss a high surface area and tunable porious structure, making them useful in applications such as gas storage, separations, and catalysis. They can range in size from microscopic crystals to bulk materials.

### 3.2.2   Property types

**Electrical properties** refer to the characteristics of a material related to its ability to conduct or resist the flow of electric current. These properties include conductivity, resistivity, and dielectric constant, which determine how well a material can conduct or insulate against electrical charges.

**Mechanical properties** describe how a material behaves under applied forces or loads. These properties include strength, stiffness, ductility, toughness, and elasticity. They determine how the material responds to stress, strain, and deformation, and are essential in understanding its structural integrity and performance.

**Stability** refers to a material's ability to maintain its properties and resist degradation over time. It encompasses chemical stability (resistance to chemical reactions or corrosion), thermal stability (ability to withstand high temperatures), and mechanical stability (ability to resist physical changes or mechanical stress).

**Optical properties** pertain to a material's interaction with light. These properties include absorption, reflection, transmission, and emission of light. Optical properties determine a material's color, transparency, opacity, and light-emitting capabilities, and are crucial in fields such as optics, photonics, and display technologies.

**Thermal properties** describe how a material conducts, stores, and dissipates heat. These properties include thermal conductivity, specific heat capacity, thermal expansion coefficient, and thermal diffusivity. Thermal properties influence a material's ability to transfer heat, its response to temperature changes, and its behavior in thermal management applications.

**Energetic properties** are computational models that use machine learning algorithms to predict the behavior of materials at the atomic or molecular level. They employ large datasets to learn the relationships between atomic arrangements and energies, enabling the simulation and understanding of complex material systems.

**Semiconductor properties** refer to the electrical behavior of materials that exhibit an intermediate conductivity between conductors and insulators. These materials can be controlled to selectively allow or impede the flow of electrons, making them ideal for electronic devices. Semiconductor properties are characterized by parameters such as band gap, carrier mobility, and doping concentration, and are crucial in the design and functionality of transistors, diodes, and integrated circuits.

### 3.2.3   Datasets

**Materials Project (MP)** Jain et al. [2013] (license: CC-BY-4.0) is a database that curates inorganic materials with computed properties including but not limited to thermal, electrical, mechanical, etc.

**MatBench** Dunn et al. [2020] (MIT license) is a benchmark that provides a standardized framework for evaluating and comparing the performance of different machine learning models on various materials science tasks. It curates data from multiple sources with MP as a main source. However, they do not provide machine learning ready data preparation nor implemented machine learning models and workflow.

**Quantum MOF Database (QMOF)** Rosen et al. [2022] (license: CC-BY-4.0) is a comprehensive database that focuses on metal-organic frameworks (MOFs) with quantum-chemical properties. The MOFs are optimized by DFT derived from both experimental and hypothetical MOF databases.

**Organic Materials Database (OMDB)** Borysov et al. [2017] (open access but no license specified) is a repository of organic materials. The properties are calculated using DFT for crystal structures contained in the COD database (in Appendix B additional data sources).

**Joint Automated Repository for Various Integrated Simulations (JARVIS)** Choudhary et al. [2020] (license: GNU v3.0) is a database that integrates materials data from various sources, including quantum mechanical calculations, materials simulations, machine learning predictions and high-throughput databases. Our datasets DFT3D, DFT2D and EDOS-PDOS are all from JARVIS database.

**Open Catalyst (OC)** Chanussot et al. [2021] (MIT license) is a database focused on catalytic materials. It includes three tasks: Structure to Energy and Forces (S2EF), Initial Structure to Relaxed Structure (IS2RS) and Relaxed Energy (IS2RE).

**Transition Metal Quantum Materials Database (tmQM)** Balcells and Skjelstad [2020] (license: CC BY-NC 4.0) is a comprehensive database focused on transition metal-based materials. It compiles experimentally derived and computationally predicted data on the structure, composition, and electronic properties of transition metal compounds.

**Quantum Machines 9** (QM9) Ramakrishnan et al. [2014] (open access but no license specified) comprises small organic molecules up to 9 heavy atoms with 12 quantum chemical properties.

**Carbon24** Pickard [2020] (license: CC-BY-4.0) is a synthetic dataset that includes materials made up by carbon atoms but with different structures obtained by *ab initio* random structure searching.

**Perov5** Castelli et al. [2012a,b] (license: CC-BY-4.0) is a synthetic dataset that includes perovskite materials with the same structure but different compositions.

### 3.3 Machine Learning Ready Dataset Preparation

**Raw data format** The raw data format for both molecules and crystals is 3D structures and atomic types. Other features (such as angular information) can be derived from them.

**Machine learning ready data format** As explained in Sec. 3.1, a machine learning ready format for materials includes *atomic types* denote the atomic number of a given atom and are often converted to one-hot embeddings; *atomic coordinates* denote the positions of a given atom and often need to be used careful if equivariance needs to be guaranteed; *edge features* denote information attached to each edge which often include bond types, interatomic distances, etc.

**Graph construction** Three common graphs are constructed to represent the materials: *multi-graph construction* is a common way to represent materials as graphs which considers edges with repeated atoms (outside the unit cell) as multiple edges with the same atom; *line graph construction* for materials representation is first proposed in Choudhary and DeCost [2021] which a bond adjacency graph (i.e. line graph) is constructed to capture the bond and angular information.

**Data split** The test scenarios are often out-of-distribution of the training set. While previously common use data split is random split, it is crucial to develop data splits that mimic the real scenarios: *composition split* (e.g. AxBy vs AzBy) refers to splitting the dataset with same materials compositions but varying ratios; *system split* (e.g. AB, AC vs ABC) refers to splitting the dataset with unseen materials systems; *time split* refers to splitting the dataset into training, validation and test set by the date when the materials are published. Note that time split is only available for MP dataset now as the publication information for each material structure is provided in MP.

### 3.4 Evaluations

#### 3.4.1 Predictive Evaluations

The evaluation metrics for predictive tasks depend on the type of prediction label: (i) **scalar value prediction**: common evaluation metrics include $R^2$, mean absolute error, and mean squared error; (ii) **classification**: common evaluation metrics are accuracy and Area under the ROC Curve (AUC-ROC) score; (iii) **vector/tensor value prediction**: common evaluation metrics are similar to scalar value prediction, including $R^2$, mean squared error, and mean absolute error. However, the distance measurement between two vector or tensor values may need to take into account the symmetry, e.g., rotation invariance, such that the distance between the rotated crystal structure and

the original structure is zero; (iv) **distribution prediction**: common evaluation metrics include cosine similarity, KL divergence, Wasserstein distance, etc.

### 3.4.2 Generative Evaluations

Evaluating generative tasks has been a notoriously challenging problem in machine learning. The evaluation metrics can generally be divided into three categories: (i) **reconstruction**: This evaluates the performance of the generative methods in reconstructing the exact material in the training set. (ii) **basic requirement**: This assesses the minimum requirement for the generated materials, such as structure or composition validity. (iii) **distribution**: This measures whether the generative model is capable of learning the data distribution (in terms of structure, property, etc.) in the training set, and whether it can interpolate or generalize to unseen materials.

We include all three types of evaluations metrics developed by Xie et al. [2022]: *Materials match* is a reconstruction metric to check if the generated material reconstructs structure in the test set. Following Xie et al. [2022], this is done using *StructureMatcher* from *pymatgen* Ong et al. [2013] which considers the match of two materials considering invariances. *Validity* is a basic metric to check if the generated materials are valid. Following Court et al. [2020], a material structure is valid if the shortest distance between any pair of atoms is greater than 0.5Å. *Structure coverage* is a distribution metric to check if the generated material structures cover the training distribution. We follow Xie et al. [2022] to utilize the CrystalNN fingerprint Zimmermann and Jain [2020] and normalized MagPie fingerprint Ward et al. [2016] to define the structure and composition distance, respectively. *Property statistics* is a distribution metric to check if the properties of generated materials are close to those in the training dataset.

### 3.5 Oracle Functions for Generative Materials Design

In our efforts to facilitate the generative design of materials, we have established two categories of oracle functions. Drawing inspiration from oracle functions designed for drug discovery via machine learning Huang et al. [2021], we initially offer a **fingerprint (FP)-based oracle function**. This function utilizes conventional materials descriptors in tandem with classical machine learning algorithms to predict properties of interest. More specifically, we have pre-trained a random forest model for each material property prediction task across 13 different datasets as proposed by Dunn et al. [2020]. Consequently, by harnessing the pre-trained models with extracted features based on the Sine Coulomb Matrix and MagPie featurization algorithms, we can predict the properties of an input material. While the predictive accuracy of these classical, FP-based materials descriptors may not rival that of deep learning-based models, we underscore the importance of their inclusion. Their utilization enables generalization where rules apply and mitigates the risk of biasing the optimization process towards deep learning. Our second oracle function is **structure-based oracle function** which aids in selecting an appropriate substrate for a given material (film). By taking into account their respective structures, we have incorporated an oracle function that matches a film with a list of substrates. Specifically, this method analyzes the compatibility between a thin film and various potential substrates, particularly in terms of crystallographic orientation, matching area, potential strain, and elastic energy. This is achieved by loading the structural information of the film and substrates from respective files, then calculating and grouping matches based on substrate Miller indices. Each match, characterized by a minimum match area, is recorded with relevant details such as the substrate formula, orientations of the film and substrate, and, if available, elastic energy and strain. Then the most suitable matches—those with the smallest matching area—for each substrate orientation are identified. This method ultimately returns a list of all matches, providing a comprehensive overview of how well the film could potentially fit on each substrate. Details can be found in Appendix E.

## 4 Benchmarking Machine Learning Models

### 4.1 Existing approach

A burgeoning amount of machine learning models have been developed for learning molecular representations suitable for a variety of downstream tasks, especially machine learning potential and molecular property prediction Wu et al. [2018], Ramakrishnan et al. [2014], Chmiela et al. [2017].

Table 2: Representative work in modeling molecular and crystal structures.

| Method | Representation | Symmetry | Graph construction | Angular |
|---|---|---|---|---|
| CGCNN Xie and Grossman [2018] | Graph | Perm. + E(3) Inv. | Multi-graph | None |
| ALIGNN Choudhary and DeCost [2021] | Graph | Perm. + E(3) Inv. | Multi-graph + Line graph | Explicit |
| SchNet Schütt et al. [2018] | Graph | Perm. + E(3) Inv. | Multi-graph | None |
| EGNN Satorras et al. [2021] | Graph | Perm. + E(3) Equiv. | Multi-graph | Implicit |
| DimeNet++ Gasteiger et al. [2020] | Graph | Perm. + E(3) Inv. | Multi-graph | Explicit |
| GemNet Gasteiger et al. [2021] | Graph | Perm. + SE(3) Inv. | Multi-graph | Explicit |
| Equiformer Liao and Smidt | Graph | Perm. + E(3)/SE(3) Equiv. | Multi-graph | Implicit |
| LEFTNet Du et al. [2023] | Graph | Perm. + E(3)/SE(3) Equiv. | Multi-graph | Implicit |

However, most of existing work focus on molecules without periodicity. Around the same time, another branch of work motivated directly by modeling crystal structures have been developed. We implement and benchmark models from both branches to facilitate the development of new methods in realization of both directions. Specifically, we detail them below and summarize them in Table 2.

**Learning on crystal structures.** *CGCNN* Xie and Grossman [2018] is a E(3) invariant graph neural network that leverages pairwise distances as edge features. *ALIGNN* Choudhary and DeCost [2021] is an E(3) invariant graph neural network that builds an extra line graph to explicitly encode the bond angle information in addition to the original atomistic graph similar to CGCNN.

**Learning on molecular structures.** *SchNet* Schütt et al. [2018] is an E(3) invariant graph neural network that leverages pairwise distances with a continuous filter convolution to construct the message. *EGNN* Satorras et al. [2021] is an E(3) equivariant graph neural network that leverages relative positions between each pair of nodes and pairwise distances as the message function to update both invariant and equivariant features. *DimeNet++* Gasteiger et al. [2020] is an E(3) invariant graph neural network that introduces bond angles to improve expressiveness. However, it requires triplet of atom representations to model the bond angle. *GemNet* Gasteiger et al. [2021] is an SE(3) invariant graph neural network that leverages dihedral angles for better expressiveness. However, it requires learning on quadruplet representations of atoms. *Equiformer* Liao and Smidt is an SE(3)/E(3) equivariant graph transformer network. Equiformer equips previous transformers with equivariant operations such as tensor product to learn equivariant features built from irreducible representations. *LEFTNet* Du et al. [2023] is an SE(3)/E(3) equivariant graph neural network based on equivariant local frames. LEFTNet first scalarizes vector and tensor features during message passing and convert them back by tensorizing the scalars through the equivariant frames proposed in ClofNet Du et al. [2022b]. LEFTNet introduces a local structure encoding and frame transition encoding components to further improve the expressiveness.

## 4.2 Experiment Set-ups

We build on top of the Open Catalyst Project (OCP) which provides reproducible implementations of commonly used 3D graph neural networks with benchmarks on OC datasets Chanussot et al. [2021]. We further implement CGCNN, ALIGNN, EGNN, Equiformer and LEFTNet as they are not included in OCP. We test all the methods on a list of 13 representative tasks from our benchmarks with three data splits (random, composition and system). We mostly use the default hyperparameters provided in the open-source code of each method and reported them in Appendix F. As OC20 and QM9 have been largely adopted in the community, we directly take the results and report in Appendix C. Most of our experiments are conducted on single 16GB V100s while some experiments with memory-intensive models on single 80GB A100s.

## 4.3 Results and Discussions

Several observations can be gleaned from our benchmark results as shown in Table 3: (i) **performance** *(observation 1)*: despite the competitive performance of advanced equivariant graph neural networks, invariant models such as DimeNet++ and ALIGNN continue to be among the state-of-the-art methods; (ii) **efficiency** *(observation 2)*: there is a significant variation in efficiency across the benchmarked models. ALIGNN, DimeNet++, GemNet, and Equiformer, as illustrated in Table 4, have particularly slow runtimes. LEFTNet presents a desirable balance of accuracy and efficiency; (iii) **data split** *(observation 3)*: more realistic data splits indeed increase the challenge of the task, particularly the system split. However, this trend does not hold for all properties, with dielectric being an exception;

Table 3: Benchmark on materials property prediction tasks (different colors denote distinct property types: purple (electrical), yellow (stability), green (thermal), red (optical), blue (mechanical), - denotes missing results due to extremely small test set after data split). The best numbers are darkly shaded, and the second-best numbers are lightly shaded.

| Split | Methods | omdb bandgap | qmof bandgap | mp bandgap | is_metal | edos | pdos | dft2d | perovskites | e_form | phonons | dielectric | log_gvrh | log_kvrh | average ranking |
|---|---|---|---|---|---|---|---|---|---|---|---|---|---|---|---|
| Random | CGCNN | 0.3351 | 0.2906 | 0.2759 | 89.16% | 0.0108 | 0.0035 | 56.2634 | 0.0600 | 0.0426 | 62.5355 | 0.3122 | 0.0924 | 0.0731 | 5.1 |
| | ALIGNN | 0.2499 | 0.2285 | 0.2010 | 90.26% | 0.0104 | 0.0036 | 56.6110 | 0.0366 | 0.0246 | 29.4330 | 0.2827 | 0.0734 | 0.0557 | 2.5 |
| | SchNet | 0.4396 | 0.3746 | 0.3298 | 88.45% | 0.0120 | 0.0040 | 53.5080 | 0.0713 | 0.0447 | 97.6870 | 0.3295 | 0.1096 | 0.0775 | 7.3 |
| | EGNN | 0.5134 | 0.3964 | 0.3263 | 88.15% | 0.0113 | 0.0035 | 51.1220 | 0.0443 | 0.0627 | 65.3510 | 0.3004 | 0.0968 | 0.0735 | 6.2 |
| | DimeNet++ | 0.2764 | 0.2379 | 0.2062 | 90.69% | 0.0111 | 0.0036 | 48.5600 | 0.0371 | 0.0219 | 39.3630 | 0.2658 | 0.0731 | 0.0522 | 2.4 |
| | GemNet | 0.2516 | 0.2327 | 0.2014 | 89.75% | 0.0104 | 0.0034 | 52.5870 | 0.0411 | 0.0236 | 48.1810 | 0.2985 | 0.0856 | 0.0555 | 2.9 |
| | Equiformer | 0.3900 | 0.3221 | 0.3050 | 88.23% | 0.0123 | 0.0039 | 53.4700 | 0.0653 | 0.0568 | 64.7630 | 0.2942 | 0.0983 | 0.0671 | 6.2 |
| | LEFTNet | 0.3143 | 0.2328 | 0.1839 | 89.96% | 0.0110 | 0.0038 | 49.3590 | 0.0398 | 0.0256 | 38.0120 | 0.3030 | 0.0792 | 0.0529 | 3.3 |
| Composition | CGCNN | 0.3516 | 0.2965 | 0.2840 | 88.20% | 0.0111 | 0.0031 | - | - | 0.0411 | - | 0.3665 | 0.0946 | 0.0610 | 5 |
| | ALIGNN | 0.2631 | 0.2266 | 0.2139 | 89.00% | 0.0108 | 0.0031 | - | - | 0.0249 | - | 0.3533 | 0.0761 | 0.0488 | 2.3 |
| | SchNet | 0.4624 | 0.3670 | 0.3236 | 87.75% | 0.0121 | 0.0034 | - | - | 0.0432 | - | 0.3792 | 0.1029 | 0.0600 | 6.9 |
| | EGNN | 0.5177 | 0.3831 | 0.3433 | 87.31% | 0.0117 | 0.0030 | - | - | 0.0625 | - | 0.3622 | 0.1070 | 0.0685 | 7 |
| | DimeNet++ | 0.2793 | 0.2339 | 0.2234 | 89.37% | 0.0116 | 0.0032 | - | - | 0.0224 | - | 0.3303 | 0.0762 | 0.0413 | 2.7 |
| | GemNet | 0.2637 | 0.2321 | 0.2203 | 88.33% | 0.0110 | 0.0029 | - | - | 0.0245 | - | 0.3279 | 0.0807 | 0.0417 | 2.4 |
| | Equiformer | 0.4050 | 0.3350 | 0.3014 | 87.66% | 0.0128 | 0.0032 | - | - | 0.0527 | - | 0.3495 | 0.0938 | 0.0552 | 5.8 |
| | LEFTNet | 0.3822 | 0.2299 | 0.2062 | 88.79% | 0.0116 | 0.0033 | - | - | 0.0276 | - | 0.3515 | 0.0797 | 0.0423 | 3.6 |
| System | CGCNN | - | 0.4224 | 0.6021 | 78.71% | 0.0127 | 0.0040 | 53.0300 | 0.1132 | 0.0557 | 142.5260 | 0.1593 | 0.1124 | 0.0966 | 5.3 |
| | ALIGNN | - | 0.3431 | 0.4997 | 80.07% | 0.0125 | 0.0041 | 43.7480 | 0.0821 | 0.0331 | 48.4010 | 0.1636 | 0.0890 | 0.0766 | 3.3 |
| | SchNet | - | 0.4863 | 0.6562 | 76.59% | 0.0131 | 0.0044 | 41.1090 | 0.1405 | 0.0512 | 196.0570 | 0.1571 | 0.1194 | 0.0971 | 6.8 |
| | EGNN | - | 0.4923 | 0.7350 | 75.04% | 0.0130 | 0.0038 | 30.8940 | 0.0981 | 0.0826 | 151.5430 | 0.1495 | 0.1181 | 0.0974 | 5.8 |
| | DimeNet++ | - | 0.3419 | 0.5086 | 80.87% | 0.0130 | 0.0041 | 35.6240 | 0.0806 | 0.0294 | 87.5960 | 0.1128 | 0.0893 | 0.0698 | 2.3 |
| | GemNet | - | 0.3412 | 0.5676 | 78.47% | 0.0122 | 0.0039 | 34.8430 | 0.0851 | 0.0422 | 113.2400 | 0.1383 | 0.0951 | 0.0712 | 3.2 |
| | Equiformer | - | 0.4272 | 0.6381 | 75.02% | 0.0136 | 0.0043 | 36.7680 | 0.0955 | 0.0603 | 168.2930 | 0.1375 | 0.1066 | 0.0841 | 5.9 |
| | LEFTNet | - | 0.3468 | 0.4550 | 78.01% | 0.0134 | 0.0043 | 34.0180 | 0.0790 | 0.0288 | 89.1200 | 0.1277 | 0.0915 | 0.0710 | 3.1 |

Table 4: Benchmark the efficiency of machine learning models with materials in different sizes (pdos∼10, e_form∼30, qmof∼100) on a single V100 GPU (each row with same batch size except when exceeding the maximum memory, running time for 10 epochs).

| | CGCNN | ALIGNN | SchNet | EGNN | DimeNet++ | GemNet | Equiformer | LEFTNet |
|---|---|---|---|---|---|---|---|---|
| pdos | 68s | 623s | 77s | 87s | 158s | 203s | 713s | 117s |
| e_form | 8572s | 41343s | 10589s | 12591s | 35622s | 40801s | 62344s | 13797s |
| qmof bandgap | 678s | 2277s | 512s | 1336s | 5240s | 4572s | 27884s | 2405s |
| average ranking | 1.33 | 6 | 1.67 | 3 | 5.67 | 6 | 8 | 4.33 |

(iv) **material type** *(observation 4)*: the performance trends across various models remain consistent for a given material property. For instance, for the bandgap property, organic crystals (OMDB) demonstrate the smallest values, followed by metal-organic frameworks (QMOF), while inorganic bulk materials (MP) exhibit the largest values.

# 5   Conclusion, Limitation and Future Outlook

In this paper, we introduce $M^2$Hub as a comprehensive platform for machine learning development in materials discovery. $M^2$Hub is a toolkit that consists of problem formulation, data downloading, data processing, machine learning methods implementations, machine learning training and evaluation procedures, and benchmark results. We cover not only the commonly considered predictive tasks on materials but also provides tools to enable the study of generative tasks on materials. Specifically, we curate 9 datasets constructed by 6 types of materials with 65 tasks across 8 property types for the predictive task. We further provide 2 synthetic datasets for the purpose of generative tasks on materials. We design 3 extra challenging and realistic data split schemes in addition to previously used random split. We believe $M^2$Hub will serve as an essential role in machine learning for materials discovery with datasets, infrastructures and benchmarks.

**Limitation**: Despite we formulate the materials discovery pipeline in the machine learning language supported by datasets, infrastructures and benchmarks, most of the tasks do not involve experiments. However, in reality, experiment is the golden standard to test new materials. It remains a challenge to develop datasets and benchmarks for machine learning models to grow in assisting the experiment phase of materials discovery such as phase demixing and experiment planning (related work is summarized in Appendix A).

**Future work**: There are multiple future directions to extend $M^2$Hub is to improve the usability for materials science community (similar to previous work Ward et al. [2018], Jacobs et al. [2020]), e.g. collect pre-trained models Xia et al., Wang et al. [2021a,b], benchmark machine learning models on specific tasks Kong et al. [2022], Bai et al. [2023], etc.

**Maintenance**: We are committed to maintaining and extending the usability of this toolkit to wider machine learning and materials science community. We plan to maintain this toolkit by adding new datasets, new tasks and evaluations, new oracle functions, and pre-trained models as this community continues to grow to benefit more researchers.

# 6 Acknowledgement

This project is partially supported by the Eric and Wendy Schmidt AI in Science Postdoctoral Fellowship, a Schmidt Futures program; the National Science Foundation (NSF), the Air Force Office of Scientific Research (AFOSR); the Department of Energy; and the Toyota Research Institute (TRI).

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

# Appendix for M$^2$Hub

## Contents

## A  Additional Related Work

As mentioned in Sec. 5, our main focus in the current version is virtual screening, inverse design and molecular simulation while ignoring tasks related to experiments where problem formulation, dataset curation, evaluation and benchmarking are much more challenging. In this section, we introduce additional related work that leverage machine learning to assist in the experiment phase of materials discovery.

### A.1  Materials Synthesis

In addition to designing materials computationally, new materials have to be synthesized with experiments. Even though computational methods have been deployed to simulate or predict material properties, it is much harder to predict how materials can be synthesized. Thus, it is a nontrivial problem to study material synthesis. Moreover, the synthesis process is challenging to predict as well. Material scientists have to post-process the synthesized material to identify the synthesized structures. Specifically, X-ray Diffraction (XRD) is a commonly used technology to detect crystal structures. Chen et al. [2021] combines deep learning with reasoning module to incorporate physical constraints and identify crystal structure phase compositions from the experimental results with XRD patterns.

### A.2  Experiment Control with Machine Learning

Experiment is an indispensable step to validate the properties of the materials. However, experiments are usually very expensive and even inaccessible in real scenarios. In addition, current experiment highly relies on human expert design which may be suboptimal. Therefore, automated experiment design becomes an urgent yet challenging problem. One way to improve the efficiency of experiment design is to leverage machine learning models for uncertainty estimation. Specifically, active learning can be utilized to construct a loop of decision and feedback. Machine learning models suggest the next experiment and the experiment provides feedback to improve machine learning models Ament et al. [2021]. Another promising direction is to leverage reinforcement learning methods which have an agent attempting to achieve some goal by the feedback provided by the environment Degrave et al. [2022].

## B  Additional Data Sources

In addition to the datasets included in our current version (Sec. 3), there are other available data sources which may be used in different purposes and we will consider to add in our future version.

- Materials Platform for Data Science (MPDS) Blokhin and Villars [2019] (no license) is a data repository that collects experimental and computational materials data through data mining from the scientific publications. There, around half a million articles were manually processed and systematized, covering a broad spectrum of physical sciences, such as physics, chemistry, materials science, environmental science, engineering, and geology.

- Crystallography Open Database (COD) Gražulis et al. [2012] (license: CC0 1.0): COD is a open-access database containing crystallographic data on inorganic and organic compounds. It includes experimentally determined crystal structures along with associated metadata for organic, inorganic, metal-organic compounds and minerals.

- Open Quantum Materials Database (OQMD) Saal et al. [2013] (license: CC-BY 4.0): OQMD is a database that focuses on quantum-mechanical calculations of materials properties. It contains calculated data on crystal structures, electronic structures, formation energies, and other material properties for a wide range of inorganic compounds.

- Inorganic Crystal Structure Database (ICSD) Hellenbrandt [2004] (commercial): ICSD is a comprehensive database that compiles > 281,000 experimentally determined crystal structures of inorganic compounds. To ensure the high quality of structures in ICSD, a structure has to be fully characterized and passed thorough quality checks by its expert editorial team before inclusion. The information in ICSD is updated biannually.

- Cambridge Structural Database (CSD) Groom et al. [2016] (commercial): CSD contains over 1.1M accurate 3D experimentally crystalized structures with data from X-ray and neutron diffraction analyses. It contains diverse types of organic crystal structure (drug, pigment, etc.) and metal-organic crystals (transition metal complex, metal-organic framework, etc.).

- Novel Materials Discovery (NOMAD) Draxl and Scheffler [2019] (license: CC-BY 4.0): NOMAD is a data management platform for materials science data where users can share data freely. Here, NOMAD is a web-application and database that allows to centrally publish data. But you can also use the its utilities to build your own local database.

- Materials Cloud (MC) Talirz et al. [2020] (license: CC0 1.0): MC is built to enable the seamless sharing and dissemination of resources in computational materials science, offering educational, research, and archiving tools; simulation software and services; and curated and raw data. These underpin published results and empower data-based discovery, compliant with data management plans and the FAIR principles Scheffler et al. [2022]. In addition to database, MC also provides lectures for computational materials science, various visualization and simulation tools.

- AFLOW Curtarolo et al. [2012] (MIT license) Similar to MC, AFLOW is a composite platform includes materials database, search and visualization, simulation, and machine learning models.

## C  Additional Experimental Results

In addition to our benchmark results, two popular benchmarking datasets, OC20 and QM9 have been extensively tested in previous work. We directly take the experimental results from Liao and Smidt for OC20 (Table.5) and Du et al. [2023] for QM9 (Table.6) as a reference on performance of existing approaches. Note that there are two commonly used data splits for QM9 in previous literature and they are both reported.

## D  Dataset Statistics

In Table 7, we report the statistics of each dataset with number of samples and number of atoms in each material.

Table 5: Benchmark on machine learning forcefield (OC20 IS2RE test set) (results taken from Liao and Smidt). (The best results are **bolded**.)

| | Energy MAE | | | | | EwT | | | | |
| Methods | ID | OOD Ads | OOD Cat | OOD Both | Average | ID | OOD Ads | OOD Cat | OOD Both | Average |
|---|---|---|---|---|---|---|---|---|---|---|
| CGCNN | 0.6149 | 0.9155 | 0.6219 | 0.8511 | 0.7509 | 3.40 | 1.93 | 3.10 | 2.00 | 2.61 |
| SchNet | 0.6387 | 0.7342 | 0.6616 | 0.7037 | 0.6846 | 2.96 | 2.33 | 2.94 | 2.21 | 2.61 |
| DimeNet++ | 0.5621 | 0.7252 | 0.5756 | 0.6613 | 0.6311 | 4.25 | 2.07 | 4.10 | 2.41 | 3.21 |
| Equiformer | **0.5037** | **0.6881** | **0.5213** | **0.6301** | **0.5858** | **5.14** | **2.41** | **4.67** | **2.69** | **3.73** |

Table 6: Benchmark on molecular property prediction (QM9) (results taken from Du et al. [2023]). (The best results are **bolded**.)

| Task | $\alpha$ | $\Delta\varepsilon$ | $\varepsilon_{\text{HOMO}}$ | $\varepsilon_{\text{LUMO}}$ | $\mu$ | $C_\nu$ | $G$ | $H$ | $R^2$ | $U$ | $U_0$ | ZPVE |
| Units | bohr$^3$ | meV | meV | meV | D | cal/mol K | meV | meV | bohr$^3$ | meV | meV | meV |
|---|---|---|---|---|---|---|---|---|---|---|---|---|
| EGNN | .071 | 48 | 29 | 25 | .029 | .031 | 12 | 12 | **.106** | 12 | 11 | 1.55 |
| Equiformer | .056 | **33** | **17** | **16** | .014 | .025 | 10 | 10 | .227 | 11 | 10 | 1.32 |
| LEFTNet | **.048** | 40 | 24 | 18 | **.012** | **.023** | **7** | **6** | .109 | **7** | **6** | 1.33 |
| SchNet | .235 | 63 | 41 | 34 | .033 | .033 | 14 | 14 | .073 | 19 | 14 | 1.70 |
| DimeNet++ | .044 | 33 | 25 | 20 | .030 | .023 | 8 | 7 | .331 | 6 | 6 | 1.21 |
| LEFTNet | .039 | 39 | **23** | **18** | **.011** | .022 | **6** | **5** | .094 | **5** | **5** | 1.19 |

Table 7: Dataset statistics (number of samples in each dataset and size of systems in each dataset.

| Datasets | dft2d | edos | pdos | qmof bandgap | omdb bandgap | dielectric | log_gvrh | log_kvrh | e_form | mp bandgap | is_metal | perovskites | phonons |
|---|---|---|---|---|---|---|---|---|---|---|---|---|---|
| number of samples | 636 | 55,659 | 14,244 | 20,425 | 12,500 | 4,764 | 10,987 | 10,987 | 132,752 | 106,113 | 106,113 | 18,928 | 1,265 |
| number of atoms | 7.19± 4.35 | 10.08±9.06 | 7.23± 5.46 | 113.67±68.86 | 82.29± 26.55 | 16.89±14.67 | 8.63±8.66 | 8.63±8.66 | 29.15±30.1 | 30.02±29.94 | 30.02±29.94 | 5.00±0.00 | 7.63 ± 3.74 |

# E  Oracle Function Details

**FP-based oracle function**   This method generates desired properties for any specific input materials. The core concept is SCM/MagPie featurization and machine learning prediction. The oracle function first reads CIF files to extract the structures or compositions of the materials. Then, the function preprocess and prepares the appropriate data format based on the task at hand. Before training, the materials data are transformed by the Sine Coulomb Matrix and MagPie featurization algorithms to convert the raw data into a form that can be used by the machine learning model. After that, a standard machine learning pipeline is built to predict the target property for these materials. The pipeline uses random forest as the machine learning model. Finally, the method saves the predictions and returns them for user convenience.

**Structure-based oracle function**   This method is used to match a given film with a list of substrates. The function first reads the input film and substrates to get their structures. Then, the *SubstrateAnalyzer* from *pymatgen* Ong et al. [2013] is called to calculate possible matches between the film and each substrate. It finds the best matches (with the smallest matching area) for each orientation of the substrate. Finally, all the match information is stored and returned for users. The match information includes the substrate's formula, orientations of the substrate and film, the matching area, and optionally, the elastic energy and strain. This substrate matching process can be useful in thin film deposition processes, where you want to match the crystal structure of a thin film material to a substrate material to ensure good adhesion and minimize defects.

Table 8: Hyparameters for benchmarked machine learning models.

| | CGCNN | ALIGNN | SchNet | EGNN | DimeNet++ | GemNet | Equiformer | LEFTNet |
|---|---|---|---|---|---|---|---|---|
| cutoff | 6.0 | 8.0 | 6.0 | 6.0 | 6.0 | 6.0 | 6.0 | 6.0 |
| max_neighbors | N/A | 12 | N/A | N/A | N/A | 50 | 500 | N/A |
| num_layers | 5 | 3 | 6 | 4 | 3 | 3 | 6 | 4 |
| hidden_dimension | 256 | 128 | 128 | 128 | 192 | 128 | 512 | 128 |
| learning rate | 1e-4 | 1e-3 | 1e-4 | 1e-4 | 1e-4 | 5e-4 | 1e-3 | 5e-4 |
| optimizer | AdamW | AdamW | AdamW | Adam | AdamW | AdamW | AdamW | AdamW |
| scheduler | N/A | N/A | N/A | N/A | N/A | ReduceLROnPlateau | Cosine | N/A |
| training epochs | 500 | 500 | 500 | 500 | 500 | 500 | 500 | 500 |

# F   Experimental Details

## F.1   Hyperparameters

We report the general hyperparamters shared across models in Table 8. For model-specific parameters, we report in `https://github.com/yuanqidu/M2Hub/config`.

Table 9: Detailed system size statistics for each task in each dataset.

| Data Name | Task | Mean | std |
|-----------|------|------|-----|
| Carbon24 | carbon24 | 9.21 | 3.58 |
| DFT2D | jdft2d | 7.19 | 4.35 |
| DFT3D | avg_elec_mass | 8.58 | 6.70 |
| DFT3D | avg_hole_mass | 8.58 | 6.70 |
| DFT3D | bulk_modulus_kv | 6.71 | 5.11 |
| DFT3D | dfpt_piezo_max_dielectric_ionic | 9.83 | 6.41 |
| DFT3D | dfpt_piezo_max_dij | 7.99 | 4.96 |
| DFT3D | dfpt_piezo_max_eij | 9.83 | 6.41 |
| DFT3D | ehull | 10.34 | 8.84 |
| DFT3D | encut | 10.33 | 8.83 |
| DFT3D | epsx | 8.57 | 7.15 |
| DFT3D | epsy | 8.57 | 7.15 |
| DFT3D | epsz | 8.57 | 7.15 |
| DFT3D | exfoliation_energy | 9.42 | 6.79 |
| DFT3D | formation_energy_peratom | 10.34 | 8.84 |
| DFT3D | kpoint_length_unit | 10.33 | 8.83 |
| DFT3D | magmom_outcar | 10.21 | 8.72 |
| DFT3D | max_efg | 5.79 | 3.53 |
| DFT3D | mbj_bandgap | 8.01 | 6.37 |
| DFT3D | mepsx | 8.06 | 6.40 |
| DFT3D | mepsy | 8.06 | 6.40 |
| DFT3D | mepsz | 8.06 | 6.40 |
| DFT3D | n-Seebeck | 10.92 | 8.59 |
| DFT3D | n-powerfact | 10.92 | 8.59 |
| DFT3D | optb88vdw_bandgap | 10.34 | 8.84 |
| DFT3D | optb88vdw_total_energy | 10.34 | 8.84 |
| DFT3D | p-Seebeck | 10.92 | 8.59 |
| DFT3D | p-powerfact | 10.92 | 8.59 |
| DFT3D | shear_modulus_gv | 6.71 | 5.11 |
| DFT3D | slme | 10.98 | 7.28 |
| DFT3D | spillage | 7.58 | 5.74 |
| EDOS-PDOS | edos_up | 10.08 | 9.06 |
| EDOS-PDOS | pdos_elast | 7.23 | 5.46 |
| MatBench | dielectric | 16.89 | 14.67 |
| MatBench | log_gvrh | 8.63 | 8.66 |
| MatBench | log_kvrh | 8.63 | 8.66 |
| MatBench | mp_e_form | 29.15 | 30.10 |
| MatBench | mp_gap | 30.02 | 29.94 |
| MatBench | mp_is_metal | 30.02 | 29.94 |
| MatBench | perovskites | 5.00 | 0.00 |
| MatBench | phonons | 7.53 | 3.74 |
| Perov5 | perov5 | 5.00 | 0.00 |
| OC20 | S2EF | 73.25 | 30.96 |
| OC20 | IS2RE | 77.75 | 31.40 |
| OC20 | IS2RS | 77.75 | 31.40 |
| OMDB | band_gap | 82.29 | 26.55 |
| QM9 | all | 18.02 | 2.94 |
| QMOF | bandgap | 113.67 | 68.86 |
| tmQM | all | 65.99 | 27.01 |

