# OpenReview forum: "M$^2$Hub: Unlocking the Potential of Machine Learning for Materials Discovery"
_NeurIPS.cc/2023/Track/Datasets_and_Benchmarks — NeurIPS 2023 Datasets and Benchmarks Poster_

### Official Review · Reviewer_1xJA · 2023-06-22
**Initial Review**

**Rating:** 7
**Confidence:** 3

**Strengths:**

I like the variety of datasets exposed (spanning single-molecule systems, interfaces between metals and molecules, and bulk materials) and the implementation of multiple neural network architectures in a common framework is very useful for comparison when developing new methods or adapting architectures to new problem domains.

**Additional Feedback:**

In general, it might be nice to hear from the authors about whether extensibility to new datasets and architectures was a priority in design.

**Clarity:**

Some paragraphs are a bit unclear or awkwardly worded, but overall the paper is straightforward to understand.

**Correctness:**

Reporting error bars for the metrics you *are* able to run multiple times would be nice; that said, I do not believe the intent is to hold up any single model presented here as the paragon of accuracy, so qualitative trends among the models studied are probably sufficient.

**Documentation:**

The details on available linked datasets seem sufficient.

**Ethics:**

No significant concerns.

**Limitations:**

Graph neural network-based architectures are very popular and this work showcases many of them; I was wondering if it would be easy to use the framework for non-graph-based architectures, such as Allegro?

**Opportunities For Improvement:**

1. I see that several pieces of the code inherit structure from the open catalyst project. It would be good to clarify which parts of the work are new implementations and which have already been released elsewhere, and, if no major adaptations are required, to simply depend on those implementations if possible (i.e. for the models that are already in the OCP).
2. The discussion around energetic properties (line 188) seems to be introducing energy models, not the energy itself as a property; it may be nice to describe more fully for a more CS-oriented audience why energy is important for materials systems.
3. Line 271 - I'm assuming there is a typo, but perhaps I'm misunderstanding; samples should be counted as valid if their shortest distance is *greater* than 0.5 A, correct?
4. Table 3 - It would be nice to include indicators for whether larger or smaller values are better in a given column (perhaps just marking the accuracies where larger values are better).
5. Table 4 - I think it would be more clear to front-load exactly what is being presented in this table (from my understanding, the runtime for 10 epochs).

**Relation To Prior Work:**

As discussed above, more clarity with what is implemented as part of this work and which pieces were already provided by other projects (such as the open catalyst project) would be beneficial.

**Summary And Contributions:**

This paper presents a suite of benchmarks and models for evaluating and advancing modern deep learning approaches in the domain of 3D materials. Adapters for a range of different datasets and architectures are exposed with a common code base, improving the accessibility of this problem to non-domain experts.

---

> ### Author Response · Authors · 2023-08-08
> **Response to Reviewer 1xJA (1/2)**
>
> Q1: I see that several pieces of the code inherit structure from the open catalyst project. It would be good to clarify which parts of the work are new implementations and which have already been released elsewhere, and, if no major adaptations are required, to simply depend on those implementations if possible (i.e. for the models that are already in the OCP).
>
> A1: Thanks for the question. Indeed, we inherit some functionalities from OCP including the OC dataset, models implemented in OCP, and the multi-GPU training and evaluation framework. Our new implementation includes new models (CGCNN, ALIGNN, EGNN, Equiformer and LEFTNet), all the new datasets except OC20, all the new data splits, all the new evaluation tasks (classification, distribution prediction), all the new generative evaluation and oracles.
>
> Q2: The discussion around energetic properties (line 188) seems to be introducing energy models, not the energy itself as a property; it may be nice to describe more fully for a more CS-oriented audience why energy is important for materials systems.
>
> A2: Energy is one of the most fundamental properties of molecules and materials that describes the interactions among atoms and electrons. Imagine we have many atoms lying in the 3D Euclidean space, the function of ground state with respect to the location of all atoms forms a surface, called potential energy surface (PES). The local minima of this PES correspond to states that are relatively stable (i.e., exist in a reasonable time scale). The saddle point between two local minima gives the energy barrier, which describes how easy one state can transfer to the other. From these special points on PES, we can derive most of the interesting knowledge of a system (what states can they be in, how will they change from one state to another, etc.). This is why energy (or free energy if we consider temperature and pressure) is important in chemistry and materials.
>
> Q3: Line 271 - I'm assuming there is a typo, but perhaps I'm misunderstanding; samples should be counted as valid if their shortest distance is greater than 0.5 A, correct?
>
> A3: Thanks for spotting it. It is a typo and we have fixed it.
>
> Q4: Table 3 - It would be nice to include indicators for whether larger or smaller values are better in a given column (perhaps just marking the accuracies where larger values are better).
>
> A4: Thanks for the suggestion. We have just added an arrow for each column to indicate larger better or lower better.
>
> Q5: Table 4 - I think it would be more clear to front-load exactly what is being presented in this table (from my understanding, the runtime for 10 epochs).
>
> A5: Thanks for the suggestion. We have highlighted the description in the caption.
>
> Q6: Graph neural network-based architectures are very popular and this work showcases many of them; I was wondering if it would be easy to use the framework for non-graph-based architectures, such as Allegro?
>
> A6: Thanks for the question. Even though the implemented methods and data preprocessing are mostly 2D or 3D graph-based, our framework is absolutely versatile to adapt to other models. Even though graph structure is not explicitly used in Allegro, the underlying architecture still takes the same input type (node features and 3D positions for each atom), so it could definitely be applied to our framework.
>
> Q7: Reporting error bars for the metrics you are able to run multiple times would be nice; that said, I do not believe the intent is to hold up any single model presented here as the paragon of accuracy, so qualitative trends among the models studied are probably sufficient.
>
> A7: Thanks for pointing it out. One main reason why we didn’t report an error bar is that many datasets are large and some models are slow to train due to computational resource limitation. We agree that the intent is to show the trend of different model performance vs their efficiency on many tasks and different splits rather than a precise ranking of the “SOTA models”.

---

> > ### Author Response · Authors · 2023-08-08
> > **Response to Reviewer 1xJA (2/2)**
> >
> > Q8: In general, it might be nice to hear from the authors about whether extensibility to new datasets and architectures was a priority in design.
> >
> > A8: Thanks for asking this question. Extensibility is actually one of our main priorities when designing this benchmark, (also one of the reasons why we build upon OCP - we did develop another framework initially and later on transferred to OCP). It is very easy to extend to more datasets, data splits, evaluations, and models. Basically, the user can just plug in their model file with a config file for the hyperparameters then they can test on any dataset. In addition, this also provides benefits for the user to use and save a model they trained for deployment. We believe with some changes, it is also suitable for pre-training on large-scale materials databases due to the nice property inherited from OCP with good documentation and multi-GPU training. For generative tasks, we do not provide any model yet as it is much less studied, our oracle is easy to use and extend.

---

### Official Review · Reviewer_PX85 · 2023-07-24
**A good benchmark paper for Materials Discovery Domain**

**Rating:** 7
**Confidence:** 5
**Clarity:** The paper is well-organized, clearly …

**Strengths:**

In recent times we have seen rapid interest among the ML and Materials science community to develop ML models for fast and accurate property prediction of materials and new materials generation.We have witnessed many DL methods being proposed by different researchers in Top venues in ML and Material Science Community. But evaluating all the models was getting difficult because everyone was coming up with new models, conducting experiments with different datasets with different train test splits, and using different evaluation criteria. Hence a predefined benchmark set consisting of all the diverse datasets, different predictive and generative tasks, train-test data split, evaluation metrics, etc is needed to perform comparative studies and evaluate the existing as well as upcoming works. This paper makes a significant contribution toward addressing this need by introducing a comprehensive benchmark. With the availability of this benchmark, researchers can now assess and compare the performance of different models without encountering the challenges posed by inconsistent methodologies in the previous works.

**Additional Feedback:**

NA

**Correctness:**

All the claims made in the submission for the paper look correct to me and the proposed benchmark, dataset, evaluation method, and experimental design are also performed correctly.

**Documentation:**

All the details about data collection and organization of the datasets are provided in the paper.

**Limitations:**

The current limitation section is merged with future work, it may be better to clearly write the limitations.

**Opportunities For Improvement:**

This paper did not cite a few of the recent papers on material property prediction, published in some top ML venues. For the sake of completeness, these works should be cited as related work on the property prediction task :

Equivariant Networks for Crystal Structures  Sékou-Oumar Kaba, Siamak Ravanbakhsh, NIPS 2022
(Matformer) Periodic Graph Transformers for Crystal Material Property Prediction Keqiang Yan et. al. NIPS 2022
CrysGNN: Distilling pre-trained knowledge to enhance property prediction for crystalline materials. Das et.al  AAAI 2023
Efficient Approximations of Complete Interatomic Potentials for Crystal Property Prediction. Yuchao Lin et. al ICML 2023
CrysMMNet: Multimodal Representation for Crystal Property Prediction. Das et.al UAI 2023

**Relation To Prior Work:**

As mentioned in the "Opportunities For Improvement" section, the authors did not mention a few of the recent works published in top Ml venues. For the sake of completeness, they should cite these works as related work on the property prediction task.

**Summary And Contributions:**

This paper aims to develop a comprehensive platform and benchmark for recent interest in machine learning development for materials discovery and property prediction. Authors provide a toolkit named $M^2Hub$, where they integrated and connect different machine learning building blocks in the materials discovery pipeline.

---

> ### Author Response · Authors · 2023-08-08
> **Response to Reviewer PX85**
>
> Q1: This paper did not cite a few of the recent papers on material property prediction, published in some top ML venues. For the sake of completeness, these works should be cited as related work on the property prediction task :
>
> Equivariant Networks for Crystal Structures Sékou-Oumar Kaba, Siamak Ravanbakhsh, NIPS 2022 (Matformer) Periodic Graph Transformers for Crystal Material Property Prediction Keqiang Yan et. al. NIPS 2022 CrysGNN: Distilling pre-trained knowledge to enhance property prediction for crystalline materials. Das et.al AAAI 2023 Efficient Approximations of Complete Interatomic Potentials for Crystal Property Prediction. Yuchao Lin et. al ICML 2023 CrysMMNet: Multimodal Representation for Crystal Property Prediction. Das et.al UAI 2023
>
> A1: Thanks for pointing out the additional related work. We have cited them in our related work section.
>
> Q2: The current limitation section is merged with future work, it may be better to clearly write the limitations.
>
> A2: Thanks for the suggestion. We have split the limitation and future work into two paragraphs with bolded headings.

---

### Official Review · Reviewer_ouYe · 2023-07-24
**Novel and rich dataset collection, solid benchmarking**

**Rating:** 7
**Confidence:** 4
**Correctness:** Yes. The dataset and evaluation setti…
**Clarity:** The paper is well-written.

**Strengths:**

* The formulated tasks are diverse covering four different predictive tasks and three generative tasks. Data instances cover various material types and properties. This should provide a thorough and reliable evaluation to ML approaches and make their adaptation into material science production much easier.
* The work should be of interest to both the boarder machine learning and material science communities. The task formulation and dataset construction
* A concrete benchmarking based on the constructed tasks and datasets.

**Additional Feedback:**

n/a

**Documentation:**

Adding documentary and/or tutorials could help audience, especially the material community, access the toolkit more easily.

**Opportunities For Improvement:**

* Only material representations in 2D/3D graphs are considered in this work. Richer types of representation can be considered. For example, given the recent attention to LLMs, it could be beneficial to consider representations or modalities that can be easier to translate into languages, or develop tools to transfer between different types of representations.
* The out-of-distribution dataset split or construction can be considered in more tasks to better reflect performance in real-world scenarios. Currently, the OOD is considered only in the IS2RE tasks.
* Toolkit API documentation may be required for easier use.

**Relation To Prior Work:**

Yes.

**Summary And Contributions:**

The work introduces a toolkit to boost machine learning applications in material science. The toolkit formulates multiple material science-related tasks in to ML problem covering the materials discovery workflow and constructs a rich collection of datasets. Additionally, the work benchmarks existing approaches in a standardized and unified protocol.

---

> ### Author Response · Authors · 2023-08-08
> **Response to Reviewer ouYe**
>
> Q1: Only material representations in 2D/3D graphs are considered in this work. Richer types of representation can be considered. For example, given the recent attention to LLMs, it could be beneficial to consider representations or modalities that can be easier to translate into languages, or develop tools to transfer between different types of representations.
>
> A1: Thanks for this insightful question! We do agree that e.g. LLMs may be powerful in creating representations for materials. However, our main focus here is the most popular machine learning models (2D/3D graph neural networks). Given the rapid development of LLMs’ application in chemistry and materials domain, it may not be the best timing to include them in a benchmark setup. LLMs may require more exploration and validation to show their effectiveness in certain tasks despite some recent works in this direction. We are open to including them but we will leave this as future work.
>
> Q2: The out-of-distribution dataset split or construction can be considered in more tasks to better reflect performance in real-world scenarios. Currently, the OOD is considered only in the IS2RE tasks.
>
> A2: Thanks for pointing it out. Our proposed splits (system, composition and time) are OOD settings and are related to OOD settings proposed in OCP. For OC20, it has both absorbates and catalysts, therefore the OOD scenario is a bit different. Specifically, OOD is considered for three scenarios (1) unseen adsorbates, (2) unseen element compositions, and (3) unseen adsorbates and catalysts compositions. We did preserve the OOD setting for OC20. For other datasets, the main differences are (1) OC includes multiple instances of the same system, (2) OC includes both catalysts and adsorbates in one system, (3) our system split (AB, AC vs ABC) is a little bit different from OC’s (called element composition in their paper) but shares a similar idea. However, we have considered all practical OOD scenarios, where we indeed observed that they are more challenging compared to the random (in-distribution) split.
>
> Q3: Toolkit API documentation may be required for easier use.
>
> A3: Thanks for the suggestion. We have added them following your advice (see the general response).

---

### Official Review · Reviewer_GSxn · 2023-07-25
**Toolkit for material discovery**

**Rating:** 7
**Confidence:** 4
**Clarity:** yes

**Strengths:**

1. One place with various training and evaluation tasks for material design.
2. dataset separated per task
3. evaluation metrics for each of the task
4. the authors performed evaluation of the proposed tasks against the baselines

**Additional Feedback:**

The work is useful. I was not able to use, so I am not sure how it is easy to use and if it is consistent with the description.

**Correctness:**

The contribution is coherent with the description, unfortunately, I could not test all functionality and the availability of the curated dataset (data is available but not tested the use in the toolkit).

**Documentation:**

The documentation, a part from the paper, is limited to the homepage of the project. The dataset is not well documented (except in the paper and some example usage) and it is not clear is the data is permanent.

**Ethics:**

No ethical considerations seen at this stage, the authors declare the license (if present) of the used dataset.

**Limitations:**

The framework collects various information (datasets, baseline and metrics) in one place and perform evaluation on the proposed tasks.
In a sense this is a positive aspect (to merge different information and tools) on the other hand is limited in novelty.

**Opportunities For Improvement:**

1. there is no installation script neither instruction (but there is a environment file so we can have the correct environment); this prevented to test the functionalities (please create a pip script - setup.py)
2. some data are shared using "figshare.com", it is not clear to me if these are permanent link (e.g. DOI)
3. the evaluation of the generative task is not easy in general, but it seems arbitrary, I could the authors comment on alternative evaluation metrics and comment on why the used these two approaches? evaluation against random forest with lower performance
4. the framework only include training with existing dataset (that is actually fine)
5. Table 1 could be extended with additional information as the size of the atomic system
6. would be better to add a section describing how to extend the toolkit
7. I find the generative positive, the justification of the use of the random forest is not clear to me. Is the task intended to be used for alternative screening. Why do you call oracle if the performance are lower? is because the training dataset is alternative? How do you expect the model to work with the toolkit dataset? did you provide an analysis in the common properties?
8. Improve documentation and use of the toolkit in the Githup page.

**Relation To Prior Work:**

The area is rapidly growing, many works are not cited, but the paper has enough references.

**Summary And Contributions:**

The paper presents a toolkit for material design.

The toolkit includes baselines for neural model for various tasks:  property prediction (regression, classification), vector/tensor output and distribution (or generative)

The toolkit makes available in one place the dataset and the baseline models, the training and the evaluation metrics.

The authors did an effort to include evaluation metrics for all tasks.

Based on the response to my comments, I increased the score.

---

> ### Author Response · Authors · 2023-08-08
> **Response to Reviewer GSxn (1/2)**
>
> Q1: there is no installation script neither instruction (but there is a environment file so we can have the correct environment); this prevented to test the functionalities (please create a pip script - setup.py)
>
> A1: Thanks for the suggestion. We have created a pip script for installation. We have also added the installation instructions to the README.md.
>
> Q2: Some data are shared using "figshare.com", it is not clear to me if these are permanent link (e.g. DOI)
>
> A2: Thanks for pointing this out! Yes, “figshare.com” publishes data with DOIs. For example, DOI can be found for Carbon dataset https://figshare.com/articles/dataset/Carbon24/22705192 as 10.24435/MATERIALSCLOUD:2020.0026/V1
>
> Q3: The evaluation of the generative task is not easy in general, but it seems arbitrary, I could the authors comment on alternative evaluation metrics and comment on why the used these two approaches? evaluation against random forest with lower performance
>
> A3: Thanks for the great question. We do agree that our proposed evaluation metrics may be improved for the generative task. However, as we mentioned, this task has not been studied very well, therefore, following the literature on developing oracle functions for drug discovery [1, 2] and one previous representative work [3], we define our oracle function in this way. The main point is that we are trying to avoid using a neural network-based oracle as it may potentially overfit and optimizing the generator on a similar architecture may exploit the oracal function. Thus we define two oracle functions, one uses traditional ML method (random forest), and the other uses structure-based functions.
>
> [1] Olivecrona, M., Blaschke, T., Engkvist, O. and Chen, H., 2017. Molecular de-novo design through deep reinforcement learning. Journal of cheminformatics, 9(1), pp.1-14.
>
> [2] Huang, K., Fu, T., Gao, W., Zhao, Y., Roohani, Y., Leskovec, J., Coley, C.W., Xiao, C., Sun, J. and Zitnik, M., 2022. Artificial intelligence foundation for therapeutic science. Nature chemical biology, 18(10), pp.1033-1036.
>
> [3] Xie, T., Fu, X., Ganea, O.E., Barzilay, R. and Jaakkola, T.S., 2021, October. Crystal Diffusion Variational Autoencoder for Periodic Material Generation. In International Conference on Learning Representations.
>
> Q4: the framework only include training with existing dataset (that is actually fine)
>
> A4: Thanks for pointing it out. Yes, we collected different sources of materials datasets (some are commonly used some are not and we do not produce our own data), but the main goal of this toolkit is to provide a unified platform for data collection, data processing, model training and model evaluation. Our toolkit makes accessibility to datasets, models, tasks and evaluations much easy. For ML users, one could just replace any model they develop to compare with baseline methods. It is also straightforward to add new datasets, evaluations, and use existing models for materials science users. We are committed to including more datasets, models and evaluations in the future.
>
> Q5: Table 1 could be extended with additional information as the size of the atomic system
>
> A5: Thanks for the suggestion. We actually included relevant information in Table 7 Appendix D. The main reason why we didn’t include it in Table 1 is that some datasets have different tasks e.g. MatBench and each task could have a very different system scale (e.g. dft2d with a mean of 7 atoms and bandgap with a mean of 30 atoms). We do agree it is helpful to include them and we have added them in Table 1 and an extra table for size of atomic system in each dataset and each task in Appendix Table 9.
>
> Q6: would be better to add a section describing how to extend the toolkit
>
> A6: Thanks for the suggestion. We have extended our future work (Sec 5) with a maintenance plan.

---

> > ### Author Response · Authors · 2023-08-08
> > **Response to Reviewer GSxn (2/2)**
> >
> > Q7: I find the generative positive, the justification of the use of the random forest is not clear to me. Is the task intended to be used for alternative screening. Why do you call oracle if the performance are lower? is because the training dataset is alternative? How do you expect the model to work with the toolkit dataset? did you provide an analysis in the common properties?
> >
> > A7: This question is partially answered in Q3. We provide more rationale here for clarity. This task is absolutely not intended to be used for the screening task as the screening task we aim to just use the best model to predict the property of interest. However, in the generative setting, we actually do also need a good model to serve as an accurate oracle, but this is currently very much under-explored. We were inspired by how the close field - drug discovery handled this task by starting with a “generalizable” and low-cost but may not be a super accurate oracle and later one it could be replaced with e.g. general-purpose pre-trained “materials foundation models” as an oracle function. We are committed to maintaining this toolkit and will update better oracles when they are available.
> >
> > Q8:Improve documentation and use of the toolkit in the Githup page.
> >
> > A8: Thanks for the suggestion. We have added them following your advice (see the general response).

---

> > > ### Author Response · Authors · 2023-08-15
> > > **Regarding missing references**
> > >
> > > We have added references following the suggestion of Reviewer PX85 (as follows). Feel free to let us know if we are still missing any important references, we would like to add them as well.
> > >
> > > Equivariant Networks for Crystal Structures Sékou-Oumar Kaba, Siamak Ravanbakhsh, NIPS 2022 (Matformer) Periodic Graph Transformers for Crystal Material Property Prediction Keqiang Yan et. al. NIPS 2022 CrysGNN: Distilling pre-trained knowledge to enhance property prediction for crystalline materials. Das et.al AAAI 2023 Efficient Approximations of Complete Interatomic Potentials for Crystal Property Prediction. Yuchao Lin et. al ICML 2023 CrysMMNet: Multimodal Representation for Crystal Property Prediction. Das et.al UAI 2023

---

### Author Response · Authors · 2023-08-08
**General Response**

We appreciate all the reviewers for their acknowledgments of our contribution to the intersection of machine learning and materials discovery. We also thank all reviewers for the valuable comments and feedback that help us improve our work. We first address the shared concern and then reply to each reviewer with individual responses. We updated the revised paper and marked all changes in the paper in red.

* Documentation & Tutorial We added a documentation md file https://github.com/yuanqidu/M2Hub/blob/master/DOUCUMENTS.md in the GitHub repo along with Jupyter Notebook on (1) how to download data, (2) how to train and evaluate models, (3) how to use the oracle function for generative tasks. https://github.com/yuanqidu/M2Hub/blob/master/tutorials

---

### Author Response · Authors · 2023-08-15
**Thanks again for your valuable reviews and further comments?**

Dear reviewers,

Thanks again for your valuable reviews! We hope our responses and revisions have addressed your concerns. We are eager to hear your feedback and happy to provide further responses if you still have any concerns! We would also appreciate it that you could acknowledge if you are satisfied with our responses as well!

Best Regards,
Authors

---

> ### Comment · Reviewer_PX85 · 2023-08-20
> **Individual response not visble**
>
> Thanks. While I can see your general response, I am not able to see your responses to my review comments.

---

> > ### Author Response · Authors · 2023-08-20
> > **Fixed response visibility**
> >
> > Dear Reviewer,
> >
> > Thanks so much for pointing it out! We have just fixed it.
> >
> > Best,
> > Authors

---

> ### Author Response · Authors · 2023-08-28
> **Thanks again for your valuable reviews and final comments?**
>
> As the discussion period is coming to an end, we wish to take this opportunity to thank the reviewers again for their extensive feedback during this process, for raising interesting and relevant questions for the work, and for highlighting the importance of M$^2$Hub to the intersection of machine learning and materials discovery.

---

### Decision · Program_Chairs · 2023-09-22

**Decision:**

Accept (Poster)

**Comment:**

This submission presents a benchmark for material design, in particular considering 3D molecular information. All reviewers agree this is a good paper and should be accepted, and therefore I too recommend acceptance.